
**The weather of 1740, the coldest year in Central Europe in 600 years**
Stefan Brönnimann[1,2,]*, Janusz Filipiak[3], Siyu Chen[1,2], and Lucas Pfister[1,2]
*[1]Oeschger Centre for Climate Change Research, University of Bern, Bern, Switzerland*
*[2]Institute of Geography, University of Bern, Bern, Switzerland*
*[3]Department of Physical Oceanography and Climate Research, University of Gdansk, Gdansk, Poland*
*corresponding author: Stefan Brönnimann, stefan.broennimann@giub.unibe.ch
**Abstract**
The winter 1739/40 is known as one of the coldest winters in Europe since early instrumental measurements
began. Many contemporary sources discuss the cold waves and compare the winter to that of 1708/09. It is less
well known that the year 1740 remained cold until August and again in October, and that negative temperature
anomalies are also found over Eurasia and North America. The 1737/40 cold season over northern midlatitude
land areas was perhaps the coldest in 300 years, and 1740 was the coldest year in Central Europe in 600 years.
New monthly, global climate reconstructions allow addressing this momentous event in greater detail, while
daily observations and weather reconstructions give insight into the synoptic situations. Over Europe, we find
that the event was initiated by a strong Scandinavian blocking in early January, allowing the advection
continental cold air. From February until June, high pressure dominated over Ireland, arguably associated with
frequent East Atlantic blocking. This led to cold air advection from the cold northern North Atlantic. During the
summer, cyclonic weather dominated over Central Europe, associated with cold and wet air from the Atlantic.
The possible role of oceanic influences (El Niño) and external forcings (eruption of Mount Tarumae in 1739)
are discussed. While a possible El Niño event might have contributed to the winter cold spells, the East Atlantic
blocking is arguably unrelated to either El Niño or the volcanic eruption. In all, the cold year of 1740 marks one
of the strongest, arguably unforced excursions in European temperature.

**Introduction**
The winter 1739/40 is known as an extremely cold winter in Central Europe, rivalling the winter of
1708/09 as the coldest in the past several hundred years. The winter was severe across Europe,
including Switzerland (Pfister and Wanner, 2021), Poland (Filipiak et al., 2019), the British Isles
(Manley, 1957; Lamb 1967), Netherlands and Germany (Jones and Briffa, 2006) and other regions.
The winter started early, already in October 1739 and ended only in June 1740, and it is particularly
well known for frozen rivers and ice floods. Filipiak et al. (2019) reported that after unusually cold
easterly winds in mid-October 1739 at the coast of the Baltic Sea, there were very heavy snowfalls
and several waves of severe frost in November 1739, January 1740 and again in February and March,
with the most extreme conditions in January 1740. The coastal waters of the Baltic Sea and
particularly the Vistula River were frozen until mid-April with the ice thickness exceeding 50 cm.
Water from the huge amounts of snow melting in April caused a large and long-lasting flood in the
Baltic lowlands. In Ireland, the intense cold lasted for weeks, interspersed with only short break of
slight thaw (Gillespie, 1939). Potatoes and turnips were destroyed, cattle and even fish died (Dickson,
1997). Among the consequences was the Irish famine of 1740/41 (Engler et al., 2013). However, the
winter was only the start of a series of adverse weather and climate events, which led to high mortality
and high cereal prices also in Central Europe (Post, 1984). Due to the frozen rivers and long-term
shutdown of mills in Poland there was even a shortage of bread, and the administrative authorities of
many cities started to provide food, wood and means of subsidence to the poorest people (Filipiak et
al. 2019). Jones and Briffa (2006) pointed out that the entire year 1740 was cold and that it
particularly contrasted with the warm 1730s.
Reconstructions of sea-level pressure have allowed characterising the anomalies atmospheric
circulation of this specific period in a bit more detail. Jones and Briffa (2006), using hand analysed
monthly sea-level pressure fields, noted that in winter, the Icelandic Low and the Azores High were
weaker than normal and the dominant feature was a continental or Scandinavian High. Engel et al.
(2013), using sea-level pressure and 500 hPa geopotential height reconstruction of Luterbacher et al.
(2002), additionally found a strong high-pressure situation in spring 1740, resembling a negative
phase of the East Atlantic pattern and leading to cold air advection from the northwest.
It is less well known, however, that the winter 1739/40 was not only cold in Europe but also in North
America and parts of Asia. A cold season (Oct-May) temperature field reconstruction for midlatitude
(35-70° N) land areas from 1701-2020 indicates that this might have been the coldest cold season of
the last 300 years (Reichen et al. 2022). Recently, a comprehensive, global 3-dimensional climate
reconstruction was published (Valler et al., 2024) and numerous additional meteorological time series
have been digitised such that we can now study this event in more detail and on the daily scale, i.e.,
the scale of the weather events.
Here we study the weather of the year of 1740 using the new reconstructions combined with daily
meteorological series. We analyse sequence of events on monthly scale, zoom into prominent cold air
outbreaks on daily scale, and analyse role of forcings and large-scale circulation mechanisms.

**Data and Methods**
*Reconstructions*
We use the ModE-RA (Modern Era Reanalysis) family of reconstructions (Valler et al., 2024), which
provide monthly, global 3-dimensional fields back to 1421. Similar as the precursor product
EKF400v2 (Valler et al., 2022), ModE-RA is based on the offline assimilation of a large amount of
natural proxies, documentary data, and instrumental observations into an ensemble of 20 atmospheric
model simulations (ModE-Sim, Hand et al., 2023). Another product, termed ModE-RAclim, was



generated by assimilating the same observations into a sample of 100 realisations, randomly drawn
from all members and all model years of ModE-Sim. Analysing ModE-Sim and ModE-RAclim along
with ModE-RA allows to disentangle the role of forcings and observations. ModE-Sim was forced by
monthly sea-surface temperatures (Samakinwa et al., 2021, Titchner and Rayner, 2014), volcanic,
land-surface and solar forcings following the PMIP4 protocol (Jungclaus et al., 2017). It does not see
the assimilated observations but only the model boundary conditions. In contrast, ModE-RAclim does
not see the time-dependent boundary conditions, but only the observations. We performed the
analyses on the individual ensemble members, but when plotting spatial fields we show the ensemble
mean only. When plotting anomalies these were expressed relative to the 30 preceding years (1710-
39). Note that the ModE-RA data set was constructed as anomalies from a 71-yr moving average,
therefore the last three decades of the data set are less well constrained.
For comparison, we also used the reconstruction XBRWccc (Reichen et al., 2022), which provides
cold season (May-Oct) temperature field reconstructions for the northern extratropics. It is based on a
Bayesian reweighting approach of model simulations that are very similar as ModE-Sim. Only
phenological data (mostly ice phenology, i.e., the freezing and thawing dates of rivers and lakes, some
plant phenological data) are used to constrain this reconstruction.
*Meteorological series*
In this paper we work with daily meteorological time series from measurements and observations,
which were inventoried in Brönnimann et al. (2019) and compiled in Lundstad et al. (2022). These
compilations are complemented with additional series. Table 1 gives an overview of the series used
and their sources.
For some of the analyses, all segments were deseasonalized by fitting and subtracting the first two
harmonics of the annual cycle and then standardized. This allows for better comparison of series with
different numbers of observations per day and allows including series on unknown scales (such as
temperature in Berlin). Note that a unique reference period that works for all series does not exist. If
possible we used 1731-50, but several of the segments were too short (in once case slightly longer;
following in existing segment). This reference is shorter than that for ModE-RA (analyses of the two
data sets are performed separately). For the special case of Montpellier, where we have very irregular
data (but which always include the monthly minima and maxima), we proceeded in the same way for
the deseasonalizing. However, because the series consists mostly of maxima and minima, it has a
standard deviation that is ca. 1.5-2 times larger than that at other stations. Therefore, we inflated the
standardized anomalies by 1.5.



**Table 1.** Locations and sources of daily weather data used in this study, variables (Var., p = pressure, mslp = mean sea-level pressure, T = temperature, dir = wind direction, RR = precipitation, wn = weather notes), period and source

| Location | Var. | Period | Source |
|---|---|---|---|
| Haarlem | T | 1735-42 | KNMI |
| Leiden | T, p | 1740-50 | KNMI |
| London | mslp | 1731-50 | Cornes et al., 2012, 2023 |
| Montpellier | (T, p)* | 1738-48 | Lundstad et al., 2022 |
| Paris | T | 1732-57 | Rousseau 2019 |
| Versailles | wn | | Société Météorologique de France, 1866 |
| Berlin | T, p | 1738-43 | Brönnimann and Brugnara, 2023 |
| Gdansk | T, p, wn | 1740 | Filipiak et al., 2019 |
| Nuremberg | p, dir | 1732-43 | Brönnimann and Brugnara, 2023 |
| Uppsala | T, mslp | 1731-50 | Bergström and Moberg, 2002 |
| Padova | mslp, RR | 1731-50 | Camuffo and Jones 2002 |
| Bologna | T | 1731-50 | Camuffo and Jones 2002 |
| Channel | dir | 1731-50 | Barriopedro et al. 2014 |
| St. Blaise | (dir), wn | | Pfister et al. 2017 |

* pressure was only used until April 1746, morning (typically 3-8 AM) and afternoon (mostly 3 PM) were treated separately.

In addition to the instrumental series, we also consulted weather diaries and other historical sources to better characterize the weather of 1740. This includes observations from Gdansk (Filipiak et al., 2019), Berlin (Brönnimann and Brugnara, 2023), Versailles (Société Météorologique de France, 1866), and St. Blaise (from EURO-CLIMHIST, Pfister et al., 2017). Note that most of these series were assimilated in ModE-RA.

*Daily reconstructions of sea-level pressure fields*

For the analyses of daily weather, we not only used the raw data, but reconstructed daily pressure fields over Europe from the pressure observations using a simple analog approach (see also Pappert et al., 2022). For that we used the ERA5 reanalysis (Hersbach et al., 2020) from 1940-2023. We extracted sea-level pressure at the 1740 observation locations, deseasonalized and standardized the data in the same way as described above (using the entire period) and then determined, for each day in 1740, the closest analog day in ERA5 within a window of ±60 calendar days of the target day. We used the Eucledian distance as a distance measure. Once the closest analog is found, the sea-level pressure field for that day is taken as the reconstruction, without any further postprocessing.

An evaluation was performed by applying the procedure to the year 1940 within ERA5 using 1941-2023 as pool of analogs. Comparing the results against the actual fields in 1940 (Fig. S1) shows excellent correlations and a low root-mean squared error over central Europe, but a rapid detorioration towards the Southwest and Northeast.

*Index time series*

In addition to spatial analyses and analyses of the instrumental series, we also calculated time series within ModE-RA. We defined Central European temperature as the average 2 m temperature in the



region 5-25° E, 45-55° N. The index was also calculated in the CRUTEM5 data set (Osborn et al.,
2021) in order to extend the reconstruction to the present. Furthermore, we calculated indices for the
North Atlantic Oscillation (NAO) and the East Atlantic Pattern (EA). The former was defined as the
sea-level pressure difference between the locations of Lisbon and Gibraltar. For the latter, different
definitions exist. We use the sea-level pressure difference between 30° E/45° N and 20° W/55° N,
which is similar to Barneston and Livezey (1987) and denoted EA1 in the following. We also define
an index EA2 as the difference between 30° E/55° N and 20° W/55° N, which is more similar to the
definition of Wallace and Gutzler (1981). Note that in all indices, only the difference was calculated
and no standardization was used, since the standard deviation in the ModE-RA datasets changes over
time. We mostly analyse Jan-Feb for NAO and Mar-May for EA1 and EA2.
Finally, we also used a NINO3.4 index (Sep-Feb) which we calculated from ModE-RA 2 m
temperature data. For addressing the volcanic forcing, we used the estimated radiative forcings for
different volcanic eruptions as given in Sigl et al. (2015). We selected eruptions with a global forcing
stronger than -2 W m$^{-2}$. For both NINO3.4 and volcanic years, we analysed the NAO and EA indices
of the subsequent winter and spring periods. For NINO3.4 we used a correlation analyses, for
volcanic eruptions compositing.

**Results**
*Descriptions of the weather and impacts in Europe*
The low temperatures in the winter 1739/40 and the consequences are well documented across
Europe. Here we present the weather information from the three locations listed in Table 1.
Interestingly, the winter 1740 was compared with the winter of 1709, which was still in the memory
of the people at that time, in several of the sources. As an example, Fig. 1 shows an excerpt of a
weather diary led by Christine Kirch (Brönnimann and Brugnara, 2023). The text, spanning a travel
from Paris to Luxembourg, speaks of freezing wine, fountains freezing to the ground, and bursting
bridges. At several instances it compares measured temperatures with those in 1709 and finds that
1740 temperatures were even lower.
Commissaire Narbonne noted the weather in Versailles from 1709-45 (Société Météorologique de
France, 1866). According to his notes, the Seine was frozen, and public fires were lit in the streets of
Paris from 9 Jan to 9 Feb 1740 and similarly in Versailles. Severe frost is noted in January, February
and March. Low temperatures are noted throughout the year. On 7-8 October, during grape harvest,
Versailles experienced a severe frost and grapes were frozen.
According to two prominent scientists of Gdansk at the Baltic Sea coast, Northern Poland – Michael
Christian Hanov (a pioneer of systematic instrumental measurements in the city) and Gottfried Reyger
(botanist and chronicler), a winter of 1740 in Gdansk was unprecedented (Filipiak et al. 2019). Hanov



recorded the lowest temperatures between 8th and 14th January, 1740 with a minimum on the morning
of the 10th January. Further, extreme cold occurred also between 1st and 7th February, 17th and 25th
February and in a few selected days in March. Reyger compared several severe winters in the 18th
century (1709, 1729, 1740 and 1784) and pointed out that winter of 1740 was undoubtedly the coldest
one, however in 1709 the duration of severe frost was even higher. Harsh weather conditions during
winter and a late and cool spring resulted in a very late appearance of vegetation – species usually
present in early March were observed only in the last days of April. Although the ice on the Baltic Sea
and the Vistula remained longer in April 1771 and 1784 than in 1740, the flood lasting many weeks
had a significant impact on the economy in 1740. Both researchers noticed unnatural behaviour of
animals and numerous cases of animals freezing, both farm animals and wild ones. Among the
increased number of human diseases, many frostbites were noticed, but the mortality rate did not
increase noticeably. Further, Hanov pointed out an exceptionally cold May with extremely cloudy
conditions (whereas a cloudiness is usually minimum in May in the annual course), fog and snow
constantly present even at the end of the month, several frosts in June and unusual weather conditions
during summer. The harvest, delayed by a cold and wet August, took place in an exceptionally sunny
and warm September (according to Reyger it was "the best weather in the whole year"), the autumn
fruit harvest was also very good. October was cold again in Gdansk. The first snowfall occurred
already on the 5th. Hanov also reported the anomalously cold weather in selected months of 1740
(particularly in January) in other cities in Europe, i.e., Königsberg, Hamburg, Kiel, Wittenberg, the
Hague, Uppsala and Petersburg.

**Fig. 1.** Excerpt of "Kirch diary" led by Christine Kirch for 13 and 14 January 1740 (see Brönnimann and
Brugnara, 2023).

In Switzerland, a detailed weather diary is available from the vine-grower family Péter from St.
Blaise. The diary notes the very low temperature from 8-12 January, which are followed by warmer





weather. However, all of February then was described as "very cold" in St. Blaise. In Februrry and
March, water bodies were frozen and navigation stopped on Lake Biel and Lake Morat, and this
continued into April (19 Apr, parts of Lake Neuchatel were frozen). Most of March the weather diary
notes "frost". Frost impact on grapevines was reported in April and May. Snowfall was observed until
8 May (at low elevations) and 20 May (at higher elevations).
*Instrumental measurements*
For the year 1740, eight daily temperature series are available, although Montpellier is very sporadic
and Haarlem and Leiden are very close. More series would exist, but are not available in daily,
digitised format (see Brönnimann et al., 2019). As an example, Fig. 2 (top) shows the raw daily mean
temperature series from Paris and Haarlem from 1738-43. The low temperatures in the winter 1739/40
clearly stand out, and it becomes visually apparent that also the other seasons were colder than the
other years shown (the winter 1741/42 also is very cold). The winter 1739/40 began early, with low
temperatures in October and November 1739. After a warm December, temperatures then dropped in
January. Low temperatures lasted consistently until August, and October and November were again
very cold.
After deseasonalizing and standardizing the series (Fig. 2, middle), it can be seen that temperatures
were below average (1731-1750, where possible) at most stations during most of the year. Only
August and September had warm intervals. In the following we discuss several episodes (marked with
grey bars) in more detail by analysing the daily series (Fig. 3) and pressure fields (Fig. 4).
One of the most severe cold spells occurred in the first half of Jan 1740. It peaked at 10-11 Jan and
brought very low temperatures to Western Europe, up to 6 standard deviations below the mean, which
is extraordinary (Fig. 3). The cold was not so intense in the North and South, i.e., in Uppsala and
Bologna (although temperature also fell below -2 standard deviations at those locations). Temperature
remained low also during the rest of the month, with a similar pattern. Pressure was below normal in
the South and above normal in the North; the gradient in the standardized anomalies persisted during
the entire month. The distinct pressure drop in Padova on 27 Jan is suspect and could be outlier, but
also Montpellier shows a pressure drop.
In early March 1740, negative temperature anomalies were observed in the South and West, though
not nearly as strong as in the January case. All stations show a very strong pressure increase from
strong negative anomalies to very high positive anomalies that persisted for 10 days. The third cold
period, in May 1740, was less homogeneous. Again, temperatures were persistently low in Western
Europe (Paris, Leiden), only slightly below normal in Gdansk and Uppsala. Temperatures were also
low in Bologna the beginning of the month and again towards 20 May. Pressure was generally below
normal, but above in London.

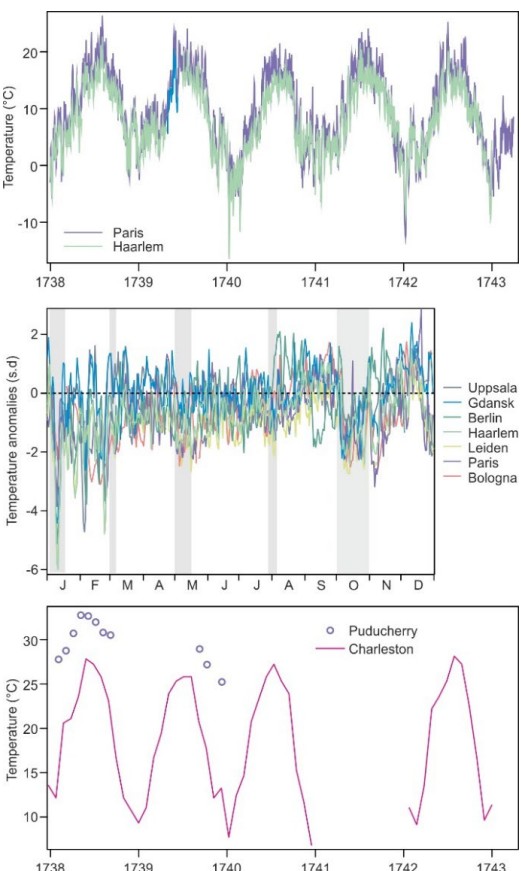


**Fig. 2.** (top) Daily temperature series from two selected European stations from 1738-43, (middle) standardised
daily temperature anomaly at seven European sites in 1740 and (bottom) the only two available non-European
temperature series that cover the boreal winter 1739/40. Shaded bars in the middle panel denote the periods
chosen for more detailed analysis.

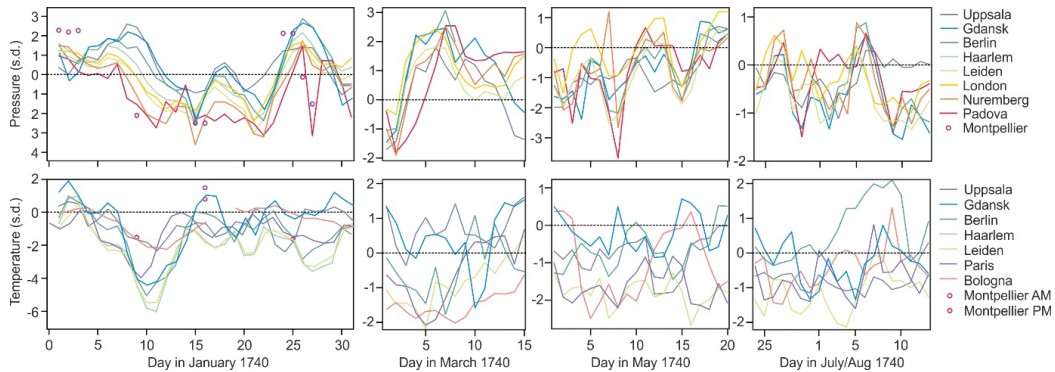


**Fig. 3.** Standardized temperature and sea-level pressure anomaly series for the four episodes 1-31 Jan, 1-15 Mar,
1-20 May, and 24 Jul to 14 Aug 1740.



The fourth chosen episode featured below normal temperature at most stations. An exception is
Berlin, where temperatures exceeded 2 standard deviations. This appears suspicious, but we have no
indications that could lead us to remove the data. Pressure was mostly below normal. Padova and
Uppsala show sometimes a different behaviour whereas all other stations run in parallel. Overall,
analysing the long pressure time series from London or Uppsala, the year 1740 did not feature
particularly many extreme days.
*Weather maps*
Plotting the daily data on a map, along with the weather observations and the analog pressure
reconstructions allows an inspection of the pressure systems and of the flow over central Europe.
During the cold spell in January (Fig. 4, top), a strong high-pressure system established over
Scandinavia, and at the same time a rather strong low pressure system developed over the northern
Mediterranean, causing a strong inverse pressure gradient across Europe. This situation can firmly be
addressed as a Scandinavian blocking event, allowing cold, continental air to flow in from the east.
The main spell lasted only five days, but further similarly extreme cold spells occurred in January and
February. In the latter cases, positive pressure anomalies were strongest over London, but stretching
into Scandinavia (not shown). Note that the sea-level pressure maps are based only on pressure
observations and are independent of temperature and wind observations.
In the first half of March, pressure was high everywhere and temperatures were below normal
everywhere except at Uppsala. Figure 4 depicts the beginning of this high-pressure period. After a
strong low-pressure situation, pressure began to build up in the West (UK) and then established over
the continent. The strongest pressure anomalies were observed first in Gdansk and Berlin. Again,
continental Europe was in an easterly flow, bringing relatively (though not extremely) cold
continental air to Central and Western Europe.
The generally low temperatures in 1740 not only included sharp but temporally limited drops of
temperature due to cold spells, but also longer, persistent phases of below normal temperature. An
example is the third selected period in May 1740. During this period, pressure was relatively low over
continental Europe and arguably higher over England. The monthly mean reconstruction shows a
strong East Atlantic pattern throughout spring. Frequent westerly or northwesterly wind arguably
brought cold air from the northern North Atlantic, which at that time of the year is very cold relative
to the land. Finally, the lowest row in Fig. 4 shows a situation in late July and early August. It was
rather cold and rainy, with typical cyclonic weather dominating. The fifth period noted in Fig. 2 is the
month of October, which was persistently cold at most stations and which will be analysed in the
following based on monthly charts.





**Fig. 4.** Standardized anomalies of pressure and temperature as well as weather observations at stations and analog sea-level pressure reconstruction (hPa) for four selected periods in Jan, Mar, May, and Jul/Aug 1740.



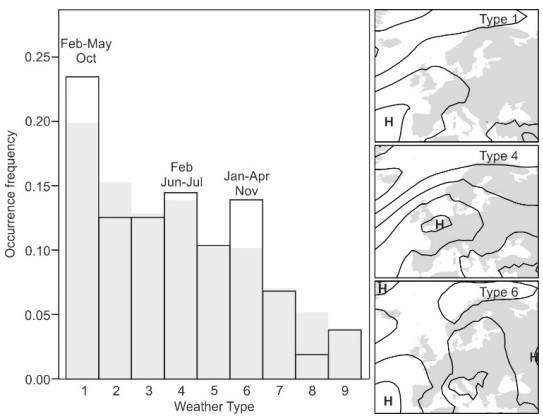

**Fig. 5.** Frequency of daily weather types in the CAP9 classification in 1740 (open rectangles) and in the period 1991-2020 (grey). Right insets show the composite fields for sea-level pressure for types 1, 4, and 6, respectively, in 1940-2020 from ERA5.

Before focusing on monthly charts, though, we would like to analyse how the daily sea-level pressure maps translate into monthly means. For this we analysed the frequency of daily weather types over central Europe, specifically the CAP9 (Cluster Analysis of Principal Components with 9 types) classification that reaches back to 1728 (Pfister et al., 2024). Three weather types were clearly overrepresented in that year, namely 1, 6, and to a lesser extent 4. These patterns (displayed in Fig. 5, right) are mostly types with high pressure systems over Western Europe.

We now turn to the analysis of monthly anomaly fields in the ModE-RA data sets (Fig. 6, see Fig. S2 for monthly anomaly fields from Oct-Dec 1739) and specifically the fields for October. Temperature anomalies in this month were negative in Central Europe. Although they were not as strong as during the winter months January to March, they reached down to -4 °C which is remarkable for this time of the year. As noted earlier, severe frost was observed in Versailles such that the grapes froze.

In ModE-RA we can also analyse monthly anomaly fields of sea-level pressure (Fig. 6, bottom, fields for Oct-Dec 1739 are shown in Fig. S2). From January into June and then again in October and November we find positive sea-level pressure anomalies in the East Atlantic and negative over Eastern Europe. This is similar to the East Atlantic Pattern, which we will address in the following. The positive anomalies could point to more frequent blocking situations. In Fig. 4 (top) we have addressed Scandinavian blocking for the cold spell in January. However, this is not seen in the monthly average, where the core of the positive anomaly is situated further in the West. The pattern resemble more a negative North Atlantic Oscillation index, although the anomaly centres are shifted southeastward.



**Fig. 6.** Monthly anomalies (with respect to 1710-39) of (top) temperature and (bottom) sea-level pressure in 1740 in the ModE-RA ensemble mean. The bottom figure also shows sea-level pressure anomalies from the analog approach (relative to 1991-2020, contour distance 2 hPa centred around zero, negative dashed).

We calculated indices for the NAO for January and February and for the East Atlantic pattern for

March to May for all three ModE products (Fig. 7, the ensemble spread is only shown for the ModE-



RA for better visualisation). In ModE-RA and ModE-RAclim, which are very similar, the NAO was
negative in 1740, but it was by no means an extreme year. However, the negative East Atlantic pattern
in spring is unique in the entire record since 1421, both for EA1 and EA2 (very similar results are
found in the annual mean). The analysis of ModE-Sim shows that only a small part of the variability
is reproduced purely from the model boundary conditions, which means that presumably the forced
component of the signal is relatively small at least in ModE-Sim. In order to extend the series to the
present we also calculated the indices in ERA5 (using 1991-2020 as a reference, correlations in the
overlapping period for NAO, EA1, and EA2 are 0.992, 0.936, 0.949, respectively). Neither of the
series shows a trend, neither in ModE-RA nor in ERA5. Also, no clear change in variability is seen in
ModE-RA, although the recent variability in the NAO in ERA5 is very large in a 600 year context.

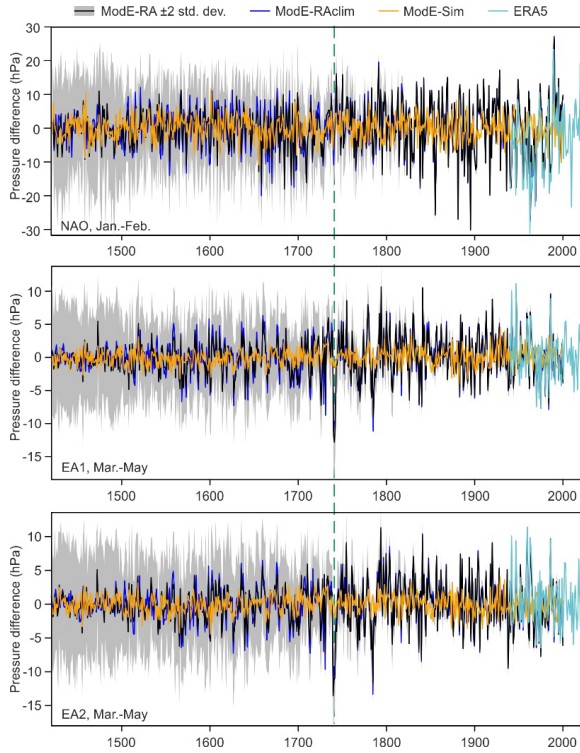

**Fig. 7.** Indices of the NAO Index in Jan-Feb and of the EA1 and EA2 in Mar-May relative to 1710-39. Shown
are the three data sets ModE-RA (grey shading denotes ±2 standard deviations of the ensemble), ModE-RAclim
and ModE-Sim as well as ERA5. The green dashed line marks the year 1740.
An interesting aspect in the monthly analysis is the persistence even at a seasonal and longer time
scale. In particular, the East Atlantic pattern is persistent or recurring. We therefore also analysed the
annual mean fields of temperature and pressure anomalies (Fig. 8). Again, ModE-RA and ModE-
RAclim show very similar patterns. For temperature, the ModE-Sim shows negative temperature
anomalies of up to 0.5 °C over parts of Europe, hence there is a contribution of boundary conditions



on a large scale, though much weaker than the full reconstruction. For sea-level pressure, there is no
contribution from ModE-Sim. The pattern in the annual mean sea-level pressure anomaly is more
similar to the East Atlantic pattern of Wallace and Gutzler (1981) rather than the corresponding
pattern in Barneston and Livezey (1987).

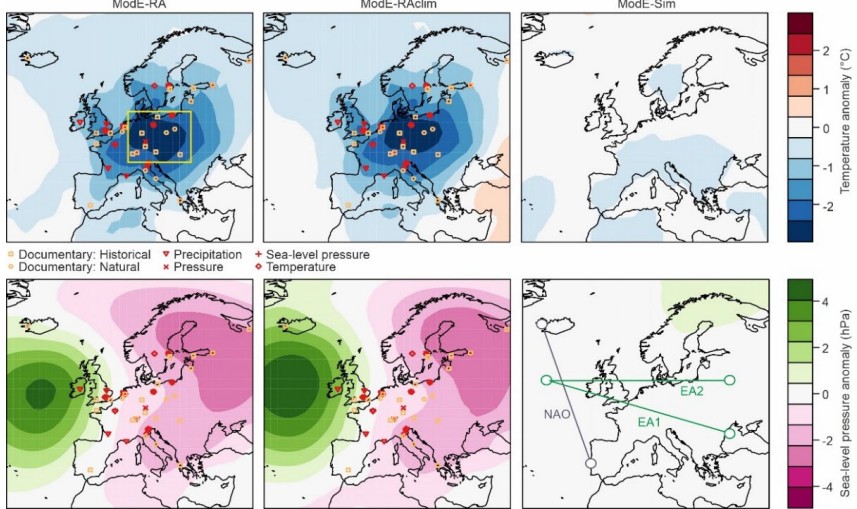


**Fig. 8.** Annual mean anomalies of (top) temperature and (bottom) sea-level pressure in 1740 in (left) ModE-RA,
(middle) ModE-RAclim, and (right) ModE-Sim. Also shown are the location and types of observations for Oct
1739-Mar 1740 on which ModE-RA and ModE-RAclim are based. The yellow rectangle(top left) shows the
region defined as Central Europe. The bottom right figure shows the definition of NAO and EA indices.

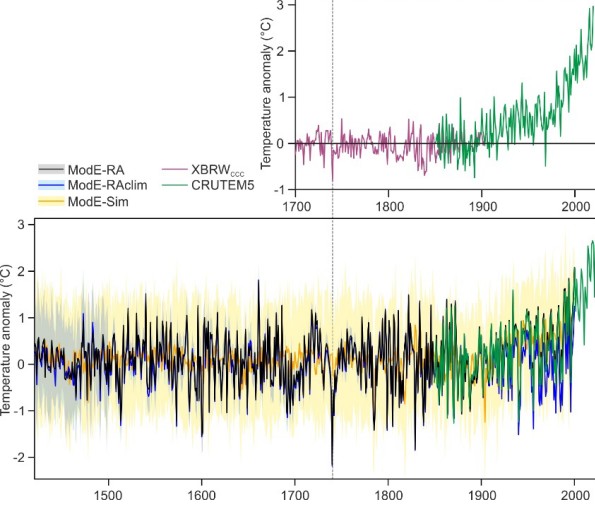


**Fig. 9.** Top: Time series of cold season (May-Oct) mean temperature over northern extratropical (35-70° N)
land areas. Bottom: Time series of annual mean, Central European temperature in the three reconstructions
ModE-RA, ModE-RAclim, and ModE-Sim. Shadings indicate two standard deviations of the ensemble.


To analyse how cold the year 1740 really was, we calculated Central European mean temperature in
the three data sets. In fact, in ModE-RA, 1740 is the coldest year on record back to 1421 (outside the
lower confidence interval of ModE-RA of any year), followed by 1829 (Fig. 9). The coldest 12-month
period (not shown) is November 1739 to October 1740. The annual mean temperature of 1740 was
2.15 °C below the preindustrial mean (1851-1900). Also shown are CRUTEM5 data in order to
extend the climate reconstructions into the present. These data show a warming of 2.5 °C since the
preindustrial, such that the cold year 1740 was more than 4 °C cooler then presently.

*A large-scale view*
The winter of 1739/40 was not only cold in Europe, but also over North America and Eurasia. This
can be seen in a recent reconstruction of cold-season (Oct-May) temperature based only on
phenological data (Fig. 10). In fact, 1739/40 was the coldest cold season in the land-area averaged
temperature between 35 and 70° N in this reconstruction (which reaches back to 1701, Reichen et al.
2022, see Fig. 9, top). The low temperatures in North America are confirmed by a temperature series
from Charleston (Fig. 2) that was not included in the reconstruction shown in Fig. 10. In fact, this is
also confirmed with documentary data. In North America, the summer of 1740 was cool and wet
(Perly, 1891). However, in ModE-RA Siberia is warmer than in XBRW$_{CCC}$.
Documentary data from China show that spring 1740 was late, both in Northern China and in
Southern China, with the end date of snow being around 20 days later than average in Beijing-
Zhangjiakou region and Nanjing (Xu, 2018; Gong et al., 1983). However, although narrative evidence
shows that the winter, especially the late winter, may have been colder than average in southern China
(Ding and Zheng, 2017; Zhang, 2004), it was not an extremely cold winter based on existing
reconstructions of East Asia (Hao et al., 2018; Wang et al., 2023).
The summer (Jun-Aug) temperature anomaly fields are very similar to those of the cold season (Fig.
10). One reason might be that for some of the rivers, the thawing takes place only shortly after the
start of the warm season assimilation window and these proxies are assimilated both for the cold and
warm season. Likewise, since the warm season assimilation window covers Apr-Sep, the tree ring
proxies in ModE-RA also affect the Oct-May period. However, the persistence might also be real as it
also appears in the analog reconstructions (contours in Fig. 6). Similar as for the cold season, Siberia
has also positive temperature anomalies in summer (arguably due to tree rings) such that the annual
mean of 1740 was not the coldest year on record in global mean temperature in ModE-RA. Sea-level
pressure anomalies show the clear EA pattern over Europe. In addition, they show a positive phase of
the Pacific North-American (PNA) pattern, most pronounced in XBRW$_{CCC}$.



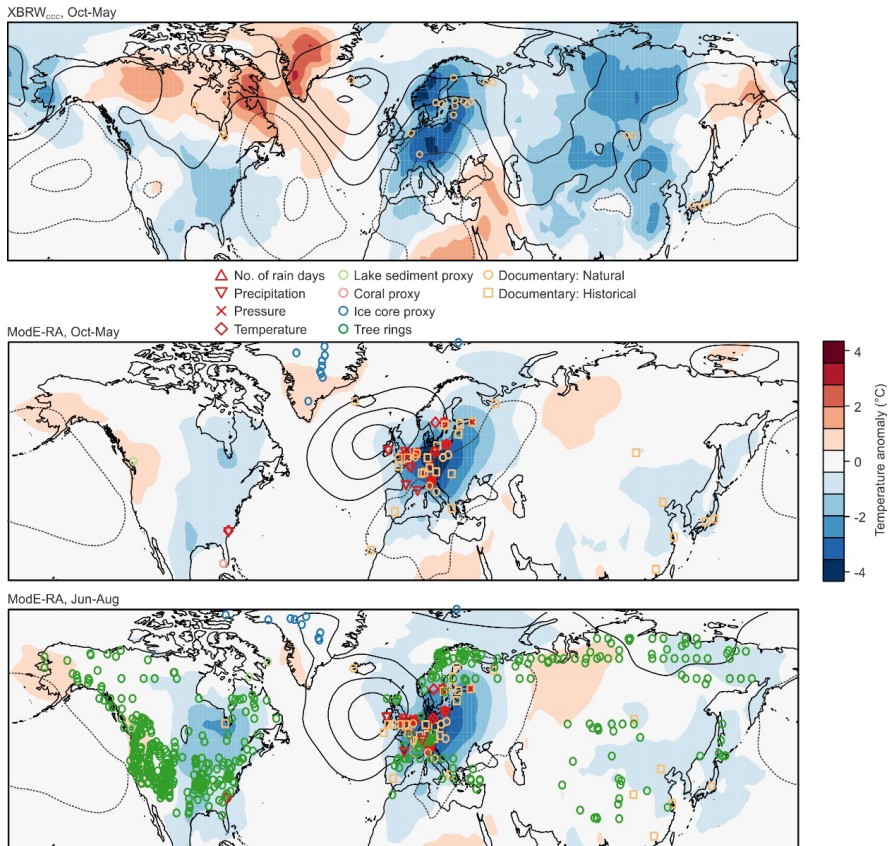


**Fig. 10.** Anomalies of temperature and sea-level pressure (contour distance 2 hPa centred around zero, negative
dashed) for (top) the cold season (Oct-May) 1739/40 in the XBRW$_{CCC}$ data set (Reichen et al., 2022), (middle)
the cold season 1739/40 in ModE-RA, and (bottom) summer (Jun-Aug) in ModE-RA, expressed as anomalies
from the preceding 30 years. For XBRW$_{CCC}$, which is based only on phenological data, orange circles mark the
locations (displayed with a slight offset if several observations, e.g., freezing and thawing dates, are available
from the same location). For ModE-RA, observations entering the data set are also shown.

*Role of forcings*

Finally, we analysed the role of oceanic influences (i.e., NINO3.4 in our case) and of external forcing

due to volcanic eruptions. ModE-RA, which is based on the monthly sea-surface temperature

reconstructions by Samakinwa et al. (2020), which in turn are based in annual reconstructions by

Neukom et al. (2019), show El Niño conditions in 1739 and partly in 1740. To analyse the possible

role of El Niño, we performed a correlation analyses, restricting our analysis to the years 1710-2000

because of the deteriorating quality further back. Results (Fig. S3) show that almost all correlations

for all ensemble members for all indices (NAO in Jan-Feb, EA1 and EA2 in Mar-May) are within

±0.1. The strongest (negative) correlations are found for the NAO. The box plots show the spread



among the ensemble members, which should not be confounded with the significance of the
correlations themselves. In fact, none of the correlations is statistically significant at p = 0.05.
Another influence could have come from the volcanic eruption of Mount Tarumae, 19-31 Aug 1739.
In the volcanic forcing data sets used in ModE-RA as well as in Sigl et al. (2015), this is not a very
big eruption, but with a global forcing of -2.4 W m$^{-2}$ exceeds the threshold set in the methods section.
We analysed all eruptions with a global forcing stronger than -2 W m$^{-2}$, again restricting ourselves to
the time period 1710-2000 (Fig. S3). We find only weak effects of the eruption, such as a slightly
positive response of the NAO in Jan-Feb and positive responses of the EA1 and EA2 pattern.

**Discussion**
*Agreement between data sets and sequence of events*
The data sets (ModE-RA and XBRW$_{CCC}$, but also ModE-RA and the analog reconstruction) agree
well with each other, demonstrating that the extremely simple analog approach is suitable for the
purpose and that it is possible to study not only climate but also the weather of 1740. Moreover, the
findings from the reconstructions are well in line with the documentary evidence.
1740 was the coldest year in central Europe since 1421 and the coldest 12-month period was Nov
1739 to Oct 1740. The cause for the cold was a specific sequence of events. It started with
Scandinavian blocking, which brought cold continental air to Central Europe. Jones and Briffa (2006)
address Jan 1740 as a continental high-pressure situation. In our data, this concerns clearly the period
5-11 January, while the monthly mean of January as a whole does not show the strongest anomalies
over Scandinavia but rather over the UK.
During spring (and actually most of the year) the dominant circulation pattern consisted of high
pressure or even blocking over the British Isles. This brought cold air from the northern North
Atlantic (which at that time of the year is much colder than the European continent) to central Europe.
August, then featured cyclonic weather, which brought cold and wet air masses form the West.
It is also important to note that the cold began already in autumn 1739 (Fig. S2) and that the following
two winters (most notably 1741-42) were also cold. Hence, a multiyear cold period followed a rather
mild decade, as pointed out by Jones and Briffa (2006).
*Dynamical aspects*
The year 1740 started with a negative NAO pattern, which however was not extreme. The cold air
outbreak in Jan 1740 is particularly noteworthy as temperature anomalies reached -6 standard
deviations. Was this the imprint of a sudden stratospheric warming (SSW)? Obviously, we have no
evidence and not even clear indications. SSWs are associated to a collapse of the polar vortex and can
affect surface weather for 30-60 days. More frequent cold air outbreaks in Northern Europe are a





possible consequence. It is not uncommon that SSWs are preceded by a pressure dipole over Europe
(Butler et al., 2017), to which Dec. 1739 bears some resemblance. Everything beyond that, however,
would be pure speculation.
Following this event, the circulation pattern over Europe took the form of a negative East Atlantic
pattern (EA1 or EA2) for a big part of the rest of the year. A similar pattern was also noted for spring
by Engel et al. (2013). In ModE-RA, the EA indices in Mar-May reached their most negative state on
record and similar for annual means. An existing reconstruction of the NAO and EA in winter
(Mellado-Cano et al., 2019), which is however based on only one series, also shows negative
anomalies in the winter 1739/40 in both indices.
In the Pacific North American sector, we find an anomaly pattern of sea-level pressure that resembles
a positive PNA phase. The relatively simple XBRW$_{CCC}$ reconstruction shows this most clearly, but it
is also seen in the ModE-RA products.
*Role of external forcings*
The role of boundary conditions (sea-surface temperatures, land surface) and external forcings can be
addressed using ModE-Sim. It shows a cooling in Central Europe of ca. 0.5 °C, i.e., a fraction of the
cooling could be due to boundary conditions. In terms of atmospheric circulation, we find a slight
negative NAO response in late winter and a very slightly negative EA pattern, but only a small part of
the deviations can be explained in that way.
In terms of external forcings, the arguably most likely candidate is the eruption of Mount Tarumae,
19-31 Aug 1739, which is incorporated in ModE-Sim. This was a highly explosive eruption (VEI=5),
but in terms of radiative forcing it was arguably not a very big eruption. It cannot be ruled out that the
eruption in the real world was larger, but there is no evidence. It can be stated that Aug 1740 was
typical for a volcanic summer, but given the location of Mount Tarumae (Hokkaido, Japan) it is not
clear whether an effect is still expected after one year. Analyses of NAO and EA indices with respect
to volcanic eruptions in general show only weak effects, which are of opposite sign to what was
observed in 1740. We therefore have no indication that the circulation anomalies in 1740 could have
been related to a volcanic eruption. Also, solar activity was average in 1740 in the PMIP4 focrcings
(Jungclaus et al., 2017).
*Role of ocean and land surface*
In the reconstructions underlying ModE-Sim, 1739/40 were El Niño years. In order to study the
possible effect of El Niño on European climate, we performed a simple correlation approach in which
we correlated NINO3.4 with indices of NAO, EA1 and EA2. We find slightly negative correlations
with NAO in Jan-Feb, which although insignificant, indicate a possible influence. In contrast, for EA1
and EA2 in Mar-May we find very small, positive correlations.



The reconstructions for 1739/40 are consistent with an El Niño winter. For instance, we see the
expected positive PNA response in the cold season 1739/40. Also the negative NAO in Jan-Feb
agrees with the correlation analysis and with the literature. El Niño events can lead to a negative,
NAO-like response (Brönnimann, 2007), to a weak stratospheric polar vortex and to more frequent
SSWs (Domeisen et al., 2019). However, other aspects do not agree. For instance, for the EA1 and
EA2 indices we find a positive correlation with NINO3.4 but strongly negative anomalies in 1740.
Furthermore, the uncertainty of El Niño reconstructions 300 years ago is high. The reconstruction by
Li et al. (2013), for instance, has no clear El Niño event.
Other teleconnection mechanisms leading to SSWs and subsequent cold air outbreaks in Europe have
been suggested in relation to recent Arctic sea ice decline. The proposed mechanism (Cohen et al.,
2014) involves an increase in snow cover over Eurasia in fall due to the low sea ice and increased
moisture transport. This could then amplify the planetary wave and lead to a collapse of the
stratospheric polar vortex. In order to test the plausibility of such a mechanism in this case we would
need to have information on sea ice or snow, which is very scattered for this period. A reconstruction
of autumn Barents-Kara Sea ice based on proxies (Zhang et al., 2018) indeed show relatively low sea
ice values (compered to the 100 years before and after) around 1740. Indications for slightly cooler
and snowy conditions are also found from other records, but they were by no means extreme (see also
Reichen et al., 2022).
In existing reconstructions, the winter 1739/40 was colder than long-term average only in South
China, and in the Yangtze River region, it was colder than past decades but not a cold winter in past
centuries (Hao et al., 2018; Hao et al., 2012). However, some of these reconstructions also confirm an
even colder winter in East Asia in 1741/42 and 1742/43. Also, the winter 1740/41 was recognized as
an extremely cold winter in southern China although not the coldest one based on narrative records
(Zheng et al., 2012). Snow cover might have provided a mechanism for the persistence of anomalies
over multiple winters (Reichen et al., 2022). However, again, this mechanism remains speculative.
*Role of atmospheric internal variability*
Finally, we have to address the role of internal atmospheric variability. In our view, after having
studied possible forcing factors and after having found no clear indications for external forcings,
oceanic or land surface effects, we ascribe most of the anomalous circulation to internal variability (in
line with interpretations by Engler et al., 2013, and Jones and Briffa, 2006). Specifically, the record
low EA1 and EA2 indices cannot be explained by any of the suggested mechanisms. These were
however, dominating the cold of the year 1740.
**Conclusions**
The year 1740 was arguably the coldest in Central Europe since 1421. The annual mean temperature
was 2 °C below pre-industrial levels, and the extended cold season 1739/40 was also the coldest one



for the northern midlatitude land mass since 1700. The winter of 1739/40 and the cold year of 1740
had severe consequences for societies in Europe, including increased prices and famine. It is therefore
relevant to assess the chain of processes causing such a cold year. Still even this large excursion of
climate dwarfs against changes observed in the last 120 years.
The analysis revealed that the coldness was due to the special sequence of events, i.e., a continental
high/Scandinavian blocking in January, then negative East Atlantic pattern during spring, a cyclonic
summer, and again negative EA pattern. Most of this is arguably due to internal atmospheric
variability. We studied many possible forcings and system effects and found no clear indications for a
forced signal. Only the circulation anomalies in January might have been made more likely by a
possible El Niño event, or, even much more speculative, low Arctic sea ice and increased snow cover.
Furthermore, part of the general cooling over Europe can be explained by a volcanic eruption in 1739.
However, this explains only a small fraction, and the most outstanding feature of this climatic
anomaly, the negative East Atlantic pattern that persisted for almost a year, shows no indication of a
forced contribution.
The analysis shows that extreme internal variability of the atmosphere is possible. It also shows that
daily weather data and a new monthly climate reconstruction together allow a detailed insight into the
mechanisms that brought forth a momentous climate event that happened close to 300 years back in
the past.

**Data availability statement:** The ModE-RA, ModE-RAclim, and ModE-Sim data (Valler et al., 2024) can be
downloaded from DKRZ (https://www.wdc-climate.de/ui/entry?acronym=ModE-RA). ERA5 reanalysis data are
available from the Copernicus Climate Change Service Data Store. XBRW$_{CCC}$ data are available from
PANGEAE (Reichen et al., 2022, https://doi.pangaea.de/10.1594/PANGAEA.934288), CRUTEM5 is available
from https://crudata.uea.ac.uk/cru/data/temperature/ (accessed 4 Mar 2024). The historical station data are
available from figshare (doi:10.6084/m9.figshare.25879186). The St. Blaise data were taken from EURO-
CLIMHIST (Pfister et al., 2017, https://www.euroclimhist.unibe.ch/, accessed 4 Mar 2024).
**Code availability statement:** All analyses were done in R using standard code. The ModE-RA family of
products can be accessed through and all corresponding analyses can also be done at the website: https://mode-
ra.unibe.ch/climeapp/.
**Author contributions:** SB performed the analyses, JF and SC provided historical observations and
documentary sources, LP provided the weather type reconstructions. All authors contributed to writing the
paper.
**Funding Information:** The work was funded by the Swiss National Science Foundation projects WeaR
(188701) and DVDW (219746) and the European Commission through H2020 (ERC Grant PALAEO-RA
787574) and the National Science Centre, Poland project No. 2020/37/B/ST10/00710.
**Competing interests.** The contact author has declared that none of the authors has any competing interests.
**Acknowledgements.** We would like to thank Yuri Brugnara, Dario Camuffo, Daniel Rousseau, Richard Cornes,
and Rolando Garcia-Herrera for providing the pressure and wind data. The simulations underlying ModE-RA
were performed at the Swiss Supercomputer Centre (CSCS).



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
