# Peer review of "The weather of 1740, the coldest year in Central Europe in 600 years"

_Climate of the Past, 2024_

## Author Comment (AC1)

**Reply to Reviewer 1**

Overview

This paper is generally well written and informative. I have a few comments below, suggesting that more could be made of the CET and the long Dutch instrumental record, and possibly the longer Dutch proxy record.

The most informative part of the paper is the determination of the very cold periods during the 1739/40 winter across Europe and the later cold spring and summer seasons. The diary information you have found links well with the few long instrumental records. Putting the diary information into monthly and seasonal context is good. Often diary information is given without this context.

You later discuss possible causes. The poor harvests might have been due to severe cold in winter, the freezing of rivers, canals and lakes, which took a long time to thaw, so planting was delayed across much of the continent.

The sea froze off the eastern UK coast in the winter of 1962/63, but there doesn't seem any mention of this for 1739/40, even though they had similar cold temperatures. 1962/63 is a more recent extremely cold winter that would be a good comparison. I see you have used winters from the early 1940s though. As an aside, the River Thames didn't freeze in 1962/63 as after London Bridge was replaced in the 1830s, the river became tidal upstream of London. From the 15th century up to bridge replacement in the 1830s the Thames wasn't that tidal above London Bridge. There is more on this in Jones (2008).

Thanks for the comment. Concerning the use of Dutch and UK data, see below. In the revised manuscript we will add a comparison to 1962/3.

Specific Comments

1. Line 32, Jones and Briffa (2006) is much more about the British Isles (maybe more so than Manley (1957) but it does provide long temperature series that are in the Netherlands and Germany. You have referred to Dickson (1997), but a couple of points about 1740 are worth mentioning: (1) It is referred to as the Forgotten Famine (the Potato famine just over a 100 years later is more well known) – you mention this a page later. Similar numbers to the 1840s left Ireland, about half to the North America and the rest to Britain. (2) During the 1739/1740 winter, the River Shannon froze over, something it's not done since. It is unsure from the Dickson book where the quite wide river did freeze over. Unfortunately there is no instrumental data from Ireland in 1740, which makes the Shannon freeze over more important.
   Thanks for this comment. In the revised manuscript we will better discuss of the Jones & Briffa paper with respect to the Irish famine and River Shannon freezing. We will also add a reference to the paper by Mateus in "Weather" for the question of instrumental data in Ireland (https://rmets.onlinelibrary.wiley.com/doi/full/10.1002/wea.3887)

2. Line 46, with respect to the warm 1730s in CET, the temperatures in this decade have only recently been exceeded (since the 1990s). The warm 1730s allowed for a dramatic increase in population of Britain and Ireland (more children survived), which might have made the impacts of 1739/40 much greater. Your later discussion doesn't seem to fully note the contrast of the cold 1739/40 compared to the very mild 1730s, especially the mild autumns in the decade.
   We will add a comment on this interesting fact here as well as later in the discussion.

3. An interesting aside about 1740 in the CET record. Using the 1961-90 base period, no year has ever had all anomalies of the same sign until 2023. Last year they were all positive. 1740 came close to being all negative, but for September.
   Thanks for this comment.

4. With Uppsala in Table 1 it is worth reading the 2002 paper in some detail. All the data before about 1739 come from a thermometer in an unheated room. If you plot the daily data, there appears to be a cut-off below which really cold temperatures in Uppsala were not measured. This does include the warm 1730s though. 1740 is the coldest year for CET, De Bilt and Berlin, but it does not appear that abnormal in Uppsala. The Q is whether this is real and was central and northern Fennoscandia not as cold as the other three locations. Uppsala would be well inside the Scandinavian High. This could be contrasted with the use of 1940 in Figure S1, which implies that northern Scandinavia may have been less cold? The less cold nature of Uppsala is mentioned later around lines 212-214.
   Thanks for this interesting point. We will add footnote to the Table and a discussion to lines 212-214

5. On line 153, you earlier referred to the winter of 1708/09. Better to do this again and to the one you're discussing as 1739/40 and later in the paragraph. Later in the paper you refer to winters by the January, which can be confusing.
   Thanks, we will be more clear by always wiring 1739/40 and 1708/09, respectively.

6. The Dutch temperature series in van Engelen et al (2001) can be used to compare 1739/40 with 1708/09 and much earlier cold winters. The series goes back to about 1250 for almost all years, and is instrumental from 1706. This series classifies 1739/40 as 8, but gives a 9 to 1788/89 and 1829/30. For these winters values come from the long De Bilt record. 1708/09 is also an 8, but 1683/84 a 9. So the Dutch series wouldn't undoubtedly say 1739/40 was the coldest, for the 18th As you point out though exactly when the coldest periods occur can have important effects on harvests and phenology, and 1740 was cold until September.
   Thanks. The Discussion section will be extended with a subsection on the comparison of this winter to other winters and on the contrast to the 1730s (see also comments above). There we will go into a bit more detail on the Dutch series.

7. The Dutch instrumental series back to 1706 is described in Labrijn (1945). I got a scanned copy of this from KNMI, and it is still available, but I can't get google to find it. It is worth getting, as it is more informative than Manley (1974). Important parts of it are summarised in English.
   Thanks. We add a brief discussion of the Dutch records. We have the De Bilt series only as monthlies, not as dailies. As all monthly series (including De Bilt) were assimilated into ModE-RA, we did not mention them specifically. In the revised manuscript we will mention the Dutch series explicitly (and add the reference) and briefly discuss the series used in ModE-RA.

8. The Thames had its greatest frost fair in 1739/40 in terms of length. I can't recall, though, the reference I read many years ago about this.
   Thanks, we will search the reference and mention this.

9. Good discussion of the circulation types in the following few pages. You note the odd values at some sites in some periods. It is quite difficult to check these, when the sites are not that close.
   Thanks

10. The temperature anomaly maps in Figure 6 seem to suggest that Uppsala wasn't as cold relatively as the other sites, but did it influence all that is further north?
    Probably the influence was large, although there are other documentary records from Stockholm and from several other locations in the Baltic sea region. We will add a brief sentence.

11. Line 336-338. You refer to 1740 as being very cold, but then mention 1829? The winter 1829/1830 was also very cold. Here you refer to winters by the December, better if you gave both years.

    Thanks, yes, we mean 1829/30, this will be corrected.

12. Just after this there are attempts to compare 1740 with respect to other years and winters, but this could also be undertaken with CET. Here 1740 was the coldest year (since 1659) at 6.9 deg C. The warmest years were 2022 and 2023, which where both 11.1 deg C, so only a 4.2 deg C difference. For CET, this is almost the same as you have in line 342, but you're talking about a west European average?

    Yes, this text is explaining Fig. 9, but we will add a comparison to CET in the discussion (new subsection mentioned above).

13. Lines 359-368, there has recently been a paper in Nature by Esper et al, which gives summer temperature estimates back 2000 years (for parts of the NH). Maybe worth comparing with that but this is summer. The coldest summer in the last 2000 years in the series was in 536, so much earlier.

    Thanks, we add this comparison to the discussion.

14. I've always wondered what might have caused 1739-40. The Japanese volcano is not large enough and the ice cores don't show a major dust/acid layer. The ENSO influence is not that strong. It could just be a natural circulation occurrence!

    We fully agree.

References

Jones, P.D., 2008: Historical Climatology – a state of the art review. *Weather* **63**, 181-186.

Labrijn, A.: The climate of the Netherlands during the last two and a half centuries, Tech. Rep. KNMI No. 102, Royal Netherlands Meteorological Institute, mededelingen en Verhandelingen, 1945. 2518

Labrijn, A.: 1945, K.N.M.I., Mededelingen en Verhandelingen no 49, Staatsuitgeverij,'s-Gravenhage.

---

## Author Response (AR1)

**Reply to Reviewer 1**

Overview

This paper is generally well written and informative. I have a few comments below, suggesting that more could be made of the CET and the long Dutch instrumental record, and possibly the longer Dutch proxy record.

The most informative part of the paper is the determination of the very cold periods during the 1739/40 winter across Europe and the later cold spring and summer seasons. The diary information you have found links well with the few long instrumental records. Putting the diary information into monthly and seasonal context is good. Often diary information is given without this context.

You later discuss possible causes. The poor harvests might have been due to severe cold in winter, the freezing of rivers, canals and lakes, which took a long time to thaw, so planting was delayed across much of the continent.

The sea froze off the eastern UK coast in the winter of 1962/63, but there doesn't seem any mention of this for 1739/40, even though they had similar cold temperatures. 1962/63 is a more recent extremely cold winter that would be a good comparison. I see you have used winters from the early 1940s though. As an aside, the River Thames didn't freeze in 1962/63 as after London Bridge was replaced in the 1830s, the river became tidal upstream of London. From the 15th century up to bridge replacement in the 1830s the Thames wasn't that tidal above London Bridge. There is more on this in Jones (2008).

Thanks for the comment. Concerning the use of Dutch and UK data, see also comment below. We added a sentence on that (l. 99/100). In the revised manuscript we also added a note on the freezing of river Thames (l. 34).

Specific Comments

1. Line 32, Jones and Briffa (2006) is much more about the British Isles (maybe more so than Manley (1957) but it does provide long temperature series that are in the Netherlands and Germany. You have referred to Dickson (1997), but a couple of points about 1740 are worth mentioning: (1) It is referred to as the Forgotten Famine (the Potato famine just over a 100 years later is more well known) – you mention this a page later. Similar numbers to the 1840s left Ireland, about half to the North America and the rest to Britain. (2) During the 1739/1740 winter, the River Shannon froze over, something it's not done since. It is unsure from the Dickson book where the quite wide river did freeze over. Unfortunately there is no instrumental data from Ireland in 1740, which makes the Shannon freeze over more important.
   Thanks for this comment. In the revised manuscript we changed this paragraph, the reference to the Jones & Briffa paper is shifted to the British Isles, and additional sentences are added for the river Thames frost fair and river Shannon freezing We also added a reference to the paper by Mateus in "Weather" for the question of instrumental data in Ireland (https://rmets.onlinelibrary.wiley.com/doi/full/10.1002/wea.3887)

2. Line 46, with respect to the warm 1730s in CET, the temperatures in this decade have only recently been exceeded (since the 1990s). The warm 1730s allowed for a dramatic increase in population of Britain and Ireland (more children survived), which might have made the impacts of 1739/40 much greater. Your later discussion doesn't seem to fully note the contrast of the cold 1739/40 compared to the very mild 1730s, especially the mild autumns in

the decade.

We added a comment on this interesting fact here as well as later in the discussion. In fact, all annual averages 1730-1738 in the Central England Temperature were above the 1961-90 average. This is now mentioned (l. 50-53).

3.  An interesting aside about 1740 in the CET record. Using the 1961-90 base period, no year has ever had all anomalies of the same sign until 2023. Last year they were all positive. 1740 came close to being all negative, but for September.

    Thanks for this comment.

4.  With Uppsala in Table 1 it is worth reading the 2002 paper in some detail. All the data before about 1739 come from a thermometer in an unheated room. If you plot the daily data, there appears to be a cut-off below which really cold temperatures in Uppsala were not measured. This does include the warm 1730s though. 1740 is the coldest year for CET, De Bilt and Berlin, but it does not appear that abnormal in Uppsala. The Q is whether this is real and was central and northern Fennoscandia not as cold as the other three locations. Uppsala would be well inside the Scandinavian High. This could be contrasted with the use of 1940 in Figure S1, which implies that northern Scandinavia may have been less cold? The less cold nature of Uppsala is mentioned later around lines 212-214.

    Thanks for this interesting point. We added footnote to the Table. 1740 is not affected by the problem of indoor vs. outdoor temperatures, but the reference period is, so if anything, we would expect the anomaly at Uppsala to be overestimated. But it is actually less extreme there than elsewhere.

5.  On line 153, you earlier referred to the winter of 1708/09. Better to do this again and to the one you're discussing as 1739/40 and later in the paragraph. Later in the paper you refer to winters by the January, which can be confusing.

    Thanks, we are more clear by always writing 1739/40 and 1708/09, respectively.

6.  The Dutch temperature series in van Engelen et al (2001) can be used to compare 1739/40 with 1708/09 and much earlier cold winters. The series goes back to about 1250 for almost all years, and is instrumental from 1706. This series classifies 1739/40 as 8, but gives a 9 to 1788/89 and 1829/30. For these winters values come from the long De Bilt record. 1708/09 is also an 8, but 1683/84 a 9. So the Dutch series wouldn't undoubtedly say 1739/40 was the coldest, for the 18[th] As you point out though exactly when the coldest periods occur can have important effects on harvests and phenology, and 1740 was cold until September.

    Thanks. The Discussion section was extended with a subsection on the comparison of this winter to other winters and on the contrast to the 1730s (see also comments above). We also refer to van Engelen (l. 445-454 are new).

7.  The Dutch instrumental series back to 1706 is described in Labrijn (1945). I got a scanned copy of this from KNMI, and it is still available, but I can't get google to find it. It is worth getting, as it is more informative than Manley (1974). Important parts of it are summarised in English.

    Thanks. We add a brief discussion of the Dutch records. We have the De Bilt series only as monthlies, not as dailies. As all monthly series (including De Bilt) were assimilated into ModE-RA, we did not mention them specifically. In the revised manuscript we mention the Dutch series explicitly and briefly discuss the series used in ModE-RA (l. 97-200).

8.  The Thames had its greatest frost fair in 1739/40 in terms of length. I can't recall, though, the reference I read many years ago about this.

    Thanks, we mention the fair in the revised manuscript (l. 34).

9.  Good discussion of the circulation types in the following few pages. You note the odd values at some sites in some periods. It is quite difficult to check these, when the sites are not that close.

    Thanks

10. The temperature anomaly maps in Figure 6 seem to suggest that Uppsala wasn't as cold relatively as the other sites, but did it influence all that is further north?
    Probably the influence was considerable, although there are other documentary records from Stockholm and from several other locations in the Baltic sea region (see also comment 4).

11. Line 336-338. You refer to 1740 as being very cold, but then mention 1829? The winter 1829/1830 was also very cold. Here you refer to winters by the December, better if you gave both years.
    Thanks, yes, we mean 1829/30, this was corrected.

12. Just after this there are attempts to compare 1740 with respect to other years and winters, but this could also be undertaken with CET. Here 1740 was the coldest year (since 1659) at 6.9 deg C. The warmest years were 2022 and 2023, which where both 11.1 deg C, so only a 4.2 deg C difference. For CET, this is almost the same as you have in line 342, but you're talking about a west European average?
    Yes, this text is explaining Fig. 9, but we add a comparison to CET in the discussion (new subsection mentioned above, specificall. L 452-453).

13. Lines 359-368, there has recently been a paper in Nature by Esper et al, which gives summer temperature estimates back 2000 years (for parts of the NH). Maybe worth comparing with that but this is summer. The coldest summer in the last 2000 years in the series was in 536, so much earlier.
    Thanks, we considered 536 to be too early to be relevant for this paper..

14. I've always wondered what might have caused 1739-40. The Japanese volcano is not large enough and the ice cores don't show a major dust/acid layer. The ENSO influence is not that strong. It could just be a natural circulation occurrence!
    We fully agree.

References

Jones, P.D., 2008: Historical Climatology – a state of the art review. *Weather* **63**, 181-186.

Labrijn, A.: The climate of the Netherlands during the last two and a half centuries, Tech. Rep. KNMI No. 102, Royal Netherlands Meteorological Institute, mededelingen en Verhandelingen, 1945. 2518

Labrijn, A.: 1945, K.N.M.I., Mededelingen en Verhandelingen no 49, Staatsuitgeverij,'s-Gravenhage.

**Reply to Reviewer 2**

General Comments

The manuscript is an interesting work on the great winter of 1739/1740 in Europe. It is well structured and raised. Text, figures and references are appropriate to the research objectives. The main conclusion is that this event was the result of natural variability, more than the consequence of radiative forcing (solar, volcanic, etc). The question here is what is the probability of this 'special sequence of events' (all of them related to atmospheric dynamics), that is, why this winter was singular, and it is very difficult to find other similar examples.

Specific comments

Table 1. I guess that 'p' is the measured pressure in each location (regardless its location above sea level), and 'mslp' is the pressure reduced to sea level. Correct?

Yes, not all series were digitized and reevaluated by us, but taken from others. These were usually taken as mslp. We added this remark to the Table caption.

Authors calculate indices for the NAO and EA patterns, but they affirm that 'during the cold spell in January a strong high pressure system established over Scandinavia and at the same time a rather strong low pressure system developed over the northern Mediterranean' (page 9, lines 245-246). This situation seems related to the positive phase of the SCAN pattern. In my opinion results would be more consistent if authors apply the same methodology used with NAO and EA to estimate the behaviour of this pattern during the studied period.

Thanks for the comment. A plot for the SCAN index is now added to the corresponding figure in the revised manuscript. It turns out that the blocking did not characterize the entire winter, so in a winter average the SCAN index is not prominent, similar as the NAO.

Figure 2 bottom. I don't find in the text comments on this figure.

It comes only very late at line 350 (of original manuscript). We moved the figure panel to the Supplement.

Figures 3 and 4. October? According to the authors 'The fifth period noted in Fig. 2 is the month of October, which was persistently cold at most stations and which will be analysed in the following based on monthly charts' (Page 9, lines 265-267). Why have you excluded October from the analysis in Figs. 3 and 4?

The paper is already relatively long and we show a lot of material. For October, we would have to show all 31 days, which would further blow up the manuscript. We therefore only analyse the monthly mean. In the revised manuscript, we add a sentence explaining this (l. 284-285).

Role of ocean and land surface (pp. 18-19). Have you considered to study the possible role of the Atlantic Multi-decadal Oscillation (AMO)? The AMO is correlated to air temperatures and rainfall over much of the Northern Hemisphere, in particular in the summer climate in North America and Europe (Ghosh et al 2016; Zampieri et al., 2017). the AMO can also modulate spring snowfall over the Alps (Zampieri et al., 2013).

We do not find a conclusive AMO signal, if anything, then the Atlantic SSTs show the classical tripole pattern, but we simply have not enough information. We now added a brief sentence with the reference to Zampieri et al., 2017.

Technical corrections

Abstract, page 1, line 15. 'The 1737/40 cold season' Erratum? Won't it be 1739/40?

Thanks

References

Ghosh et a., l2016. Impact of observed North Atlantic multidecadal variations to European summer climate: a linear baroclinic response to surface heating". Climate Dynamics. **48** (11–12): 3547. doi:10.1007/s00382-016-3283-4.

Zampieri et al., 2017. Atlantic multi-decadal oscillation influence on weather regimes over Europe and the Mediterranean in spring and summer". Global and Planetary Change. **151**: 92-100.  doi:10.1016/j.gloplacha.2016.08.014.

Zampieri et al., 2013. "Atlantic influence on spring snowfall over the Alps in the past 150 years". Environmental Research Letters. **8** (3): 034026.  doi:10.1088/1748-9326/8/3/034026.

**Reply to Reviewer 3**

The paper contains relevant information to evaluate the exceptional 1740 cold event.

It is well written and it is presented in a well-structured way, with valid scientific approach and applied methods.

It is interesting the analog pressure reconstruction to investigate the potential pressure systems that lead to the cold flow over central Europe. This strongly helps in interpreting climate anomalies in light of atmospheric circulation.

The investigation is rigorous in exploiting all the available information recovered (both at daily and longer time resolutions) and the final discussion is honestly presented, clearly underlining what robustly emerges from the analyses and giving a marginal role to purely speculative interpretations.

Please, consider that in Camuffo et al. (2017a, 2019, 2020, 2024) new versions (with respect to the Camuffo and Jones 2002 mentioned in the paper) of daily temperature and precipitation series for Bologna and Padua are presented.

Thanks. We have carefully checked our data. Indeed, our Bologna data was the new version, so the reference was wrong and we updated the reference. As to the Padua data, temperature for Padua was not included at all in our analysis. That paper was published only on 7 June this year. In the revised manuscript we now add this new series, it is very well in line with all other series. Thanks!

This leads to new versions of Figs. 2, 3, and 4, in which we now have one additional series.

Evaluate to mention that important frozen events occurred in the Venice lagoon in 1740, as documented by Gallicciolli: from Memorie Veneziane antiche profane ed ecclesiastiche of 1795 by Gallicciolli GB we read "The 6th January 1708 [More Veneto] the horrible cold started. The Lagoons were frozen over for about 18 days. Food supply was made with carriages. Similarly, in 1740, 1758 and 1788." (see Camuffo et al., 1987, 2017b). NOTE: Pay attention to "more veneto" dating (it means "according to Venetian custom", a kind of dating of the old Republic of Venice): February was the last month of the year and March the first of the new year, so that 20 February 1708 more veneto should be converted into 20 February 1709 in the Gregorian style, while 3 March 1709 more veneto remains 3 March 1709.

Thanks, this is now mentioned both in the introduction and in the Discussion (comparison with other winters).

MINOR COMMENTS:

Pag. 3, line 86: May-Oct should be Oct-May.

Thanks

Figure 9. In the caption May-Oct should be Oct-May

Thanks

Bibliography

Camuffo D. 1987. Freezing of the venetian lagoon since the 9th century a.d. in comparison to the climate of western Europe and England, Climatic Change 10 (1987) 43-66.

D Camuffo, A della Valle, F Becherini, V Zanini (2020) Three centuries of daily precipitation in Padua, Italy, 1713–2018: history, relocations, gaps, homogeneity and raw data, Climatic Change, 162, 923–942. https://doi.org/10.1007/s10584-020-02717-2

D Camuffo, F Becherini, A della Valle (2019) The Beccari series of precipitation in Bologna, Italy, from 1723 to 1765. Climatic Change, 155, 359–376. https://doi.org/10.1007/s10584-019-02482-x

Camuffo D, Bertolin C, Craievich A, Granziero R, Enzi S, (2017b) When the Lagoon was frozen over in Venice from A.D. 604 to 2012: evidence from written documentary sources, visual arts and instrumental readings. Méditerranée, Varia 1–68. http://mediterranee.revues.org/7983

Camuffo D., A. della Valle, C. Bertolin, E. Santorelli (2017a). Temperature observations in Bologna, Italy, from 1715 to 1815: a comparison with other contemporary series and an overview of three centuries of changing climate. Climatic Change, 142, 7-22. DOI 10.1007/s10584-017-1931-2

Stefanini, C.; Becherini, F.; Valle, A.d.; Camuffo, D. Homogenization of the Long Instrumental Daily-Temperature Series in Padua, Italy (1725–2023). Climate 2024, 12, 86. https://doi.org/10.3390/cli12060086